environmental chemistry/chemical engineering/ materials science

CeO2, CuO, ZrO2, H2-temperature-programmed reduction, oxygen storage capacity

**Author for correspondence:**
Nguyen The Luong
e-mail: luong.nguyenthe@hust.edu.vn

†Present address: The Department of Internal Combustion Engine, School of Transportation Engineering, Hanoi University of Science and Technology, No.1 Dai Co Viet street, Hanoi, Vietnam.

This article has been edited by the Royal Society of Chemistry, including the commissioning, peer review process and editorial aspects up to the point of acceptance.

# Structure and catalytic behaviour of CuO–CeO2 prepared by high-energy ball milling

Nguyen The Luong†, Hideyuki Okumura, Eiji Yamasue and Keiichi N. Ishihara

Department of Socio-Environmental Energy Science, Graduate School of Energy Science, Kyoto University, Yoshida Honmachi, Sakyo-ku, Kyoto 606-8501, Japan

NT, 0000-0003-3939-8266

The aim of this study is to prepare CuO–CeO2 composite by means of mechanical milling and to investigate its characteristics as a catalyst. The structural and morphological features of milled samples are observed by X-ray diffractometry and scanning electron microscopy. The redox property and total OSC (oxygen storage capacity) of the milled sample were measured by using GC-TCD and TG-DTA, which are important parameters to indicate the effectiveness of catalysts. Interestingly, reduction of CuO is repeatedly observed when milling of CuO–CeO2 powder mixture is processed in air. The redox property of milled CuO–CeO2 sample is investigated by H2-TPR, where three reduction peaks are observed for 0 h milling and only one broad peak for various other milling times. The total OSC of mechanically driven CuO–CeO2 catalyst is much higher than that of the CeO2–ZrO2 traditional catalyst system at low temperatures.

## 1. Introduction

Nowadays, more than 95% of vehicles produced are equipped with a catalytic converter [1]. The three-way catalysts (TWCs), used for the gasoline-fuelled engine are capable of simultaneously converting CO, hydrocarbon (HC) and $NO_x$, with a stoichiometric air-to-fuel ratio (A/F = 14.7), into harmless $CO_2$, $H_2O$ and $N_2$ [1]. Oxygen storage capacity (OSC) is one of the crucial factors for the performance of TWCs. The $CeO_2$–$ZrO_2$ composite is well known as an excellent promoter of OSC, where $CeO_2$ exhibits the oxygen storage/release behaviour by redox variation of Ce ions between $Ce^{3+}$ and $Ce^{4+}$, while the introduction of $ZrO_2$ into $CeO_2$ improves the reduction temperature of ceria through structural modification of the ceria lattice [2], although the OSCs at low

temperatures are still not high [1]. Many studies on $CeO_2$-based materials have been reported, such as $CeO_2$–$Al_2O_3$ [3], $CeO_2$–$SiO_2$ [4], $CeO_2$–$La_2O_3$ [3,5,6], $CeO_2$–$TbO_x$ [7], $CeO_2$–$PrO_x$ [8] and $CeO_2$–$MO_x$ (M: Zr, Ti, Cu) [9,10], to improve the OSC and increase the thermal stability.

As legislation becomes tighter, it is necessary to improve the efficiency of TWCs at low temperatures under an oxygen-rich atmosphere. The copper/copper oxides are known to exhibit oxygen storage/ release behaviour at low temperatures, although it causes fragmentation due to the large volume change [11]. Then, various metal oxides without large volume change, such as $CuMO_2$ (M = Al, Fe, Mn, Ga), have been investigated to reduce fragmentation [11], where the reduction of $Cu^{2+}$ to Cu is studied by the $H_2$-TPR (temperature-programmed reduction) and the OSC is improved at lower temperatures.

It is known that $CuO$–$CeO_2$ mixed oxides exhibit high levels of oxidation of carbon monoxide and hydrocarbon [12–15], $SO_2$ reduction by CO [16–18], NO reduction [19] and phenol oxidation [20–22]. It has also been shown that the redox properties and catalytic performance strongly depend on the preparation methods, such as sol–gel [23], hydrothermal routes [24,25], precipitation method [26], reverse micelle [27], sonochemical [28], chemical vapour deposition [29], flux method [30], micro-wave heating [31] and surfactant-assisted method [32]. The preparation condition and the mixed-oxide composition influence the phase, morphology and distribution of copper species on ceria. The enhanced catalytic activity results from interactions among the copper–cerium oxide phases.

Mechanical milling has long been used to prepare non-equilibrium materials, solid solutions and other metastable phases and also to drive mechanochemical reactions. It has been shown that, because the enhanced reaction rate can be achieved and dynamically maintained during milling as a result of microstructural refinement and mixing processes accompanying repeated fracture, deformation and welding of particles during collision events [33], several treatments employing milling could be applied for various preparation stages of mixed oxides [34–37]. Recently, the mixed oxides containing $CeO_2$ and other dopants, such as $ZrO_2$, $TbO_x$ and $HfO_2$, have been prepared by mechanical milling [7,38] with strong enhancement of the OSC properties of $CeO_2$. In addition, Castricum et al. report that the milling process of mixed Cu, $Cu_2O$ or CuO and ZnO in synthetic air results in oxidation of Cu precursors, while, under vacuum, it results in reduction. They also report that the mechanochemical reactions are promoted by mechanical milling in the presence of ZnO [39].

The OSC property of the $CuO$–$CeO_2$ system for TWCs has not been previously reported, and it is interesting to study the effect of the mechanochemical process on the OSC property of the Ce–Cu–O systems. It is also reasonable to consider that the valence change of $Ce^{4+}$/$Ce^{3+}$ and/or $Cu^{2+}$/$Cu^+$/Cu may improve the OSC property at lower temperatures. Thus, the primary aim of this study is to characterize the $CeO_2$–$CuO$ mixed oxides prepared by high-energy mechanical milling and evaluate the OSC properties, which is compared with $CeO_2$–$ZrO_2$ traditional catalysts prepared under the same experimental conditions.

# 2. Experiment

## 2.1. Catalyst preparation

Monoclinic CuO (Nilaco Corporation, less than 250 μm, 99.999% purity), monoclinic $ZrO_2$ (Nilaco Corporation, less than 200 μm, 99.8% purity) and cubic $CeO_2$ (Kojundo Chemical, less than 180 μm, 99.99% purity) powders were used as starting materials. The molar ratio of CuO in the composite was changed to be 0, 20, 30, 50, 80 and 100%. High-energy vibratory ball milling (Super-Misuni, Nissin Giken Co. Ltd.) was employed, with a rotational speed of 710 r.p.m., where the milling atmosphere was ambient. The powders and zirconia balls (ϕ10 mm) were charged in a stainless steel vial (ϕ100 mm), where the ball-to-powder weight ratio was 18:1 (18 g balls per 1 g powder) and the milling durations were changed from 0 to 30 h.

## 2.2. Characterizations

The structures, the morphological aspects and the compositions of milled samples were analysed by X-ray diffractometry (XRD) using Cu Kα radiation (RIGAKU RINT-2100CMT), scanning electron microscopy (SEM, JEOL, JSM-5800) and EDX (energy-dispersive X-ray). The silver powder (99.8% purity) was used for both the 2-theta calibration of X-ray diffraction line positions and the background intensity calibration, and the lattice parameter and the crystallite size were then calculated on the basis of FWHM (full width at half maximum intensity) of the 220 peak of $CeO_2$, where the Scherrer's Equation was used for the latter. The surface areas were estimated by the $N_2$ adsorption method (single-point BET).

## 2.3. Catalytic property measurements

The milled sample was subjected to measurement of total OSC according to the method by Tanabe *et al.* [40] and Morikawa [41]. The weight change of the milled sample was measured with TG-DTA (RIGAKU TG-8120) by the following procedure; the milled sample (about 20 mg) on an alumina container was completely oxidized at 500°C in a $N_2$-20vol%$O_2$ mixed gas flow (dry air, 500 ml min$^{-1}$) for 60 min, followed by cooling to 300°C. Then, the gas atmosphere was switched to an Ar-5vol%$H_2$ flow (500 ml min$^{-1}$) and the weight decrease due to reduction was monitored until no weight change was observed. Afterward, the gas atmosphere was again switched to the dry air and the weight increase due to oxidation was monitored until no weight change was observed. The procedure was repeated twice.

The dynamic reduction behaviour was measured by TPR, where the milled sample (about 50 mg) was put in a quartz reactor and heated at 400°C for 1 h under a $N_2$-20%$O_2$ gas flow (30 ml min$^{-1}$) and cooled to room temperature (RT). The gas was then changed to Ar-5%$H_2$ (25 ml min$^{-1}$) and the sample was heated at 15°C/min for the temperature range of 35–900°C, where the $H_2$ consumption was measured by GC-TCD (Varian CP-4900). For the second run of $H_2$-TPR, the sample (50 mg) after the first TPR was cooled to RT in Ar-5%$H_2$ gas, re-oxidized at 400°C for 1 h under a $N_2$-20%$O_2$ gas flow (30 ml min$^{-1}$), cooled to RT again, and finally heated under an Ar-5%$H_2$ gas flow (25 ml min$^{-1}$) at 15°C min$^{-1}$ up to 900°C for the repeated $H_2$-TPR.

# 3. Results and discussion

## 3.1. Structural characterization

The XRD patterns of milled samples, having a composition of 50%mol CuO and 50%mol $CeO_2$, are shown in figure 1. The reflection peak intensities of the CuO phase are largely reduced with peak broadening after 2 h milling, while no appearance of other phases is detected. Although not detected, some CuO phases may exist in nanostructured or amorphous states. After 7 h milling, the CuO peaks are almost eliminated and replaced instead by the appearance of the faint reflections from a $Cu_2O$ phase and the clear peaks of fcc Cu. The coexistence of both phases lasts until the milling duration of around 14 h. The $Cu_2O$ reflections become weaker with milling, while the Cu peaks become more distinguished with milling, a tendency which continues up to 30 h milling. There is no change in the observed phase of $CeO_2$ up to 30 h milling, although the peak broadening with milling is significant, particularly during the early milling stages regarding the change of the peak shape.

From the XRD results, the reductive valence change of CuO, i.e. $Cu^{2+} \rightarrow Cu^{1+} \rightarrow Cu$ or $Cu^{2+} \rightarrow Cu$, occurs during milling. This is consistent with a report that there are three ways [12,13] to reduce $Cu^{2+}$ to Cu: (i) $CuO \rightarrow Cu_4O_3 \rightarrow Cu_2O \rightarrow Cu$, (ii) $CuO \rightarrow Cu_2O \rightarrow Cu$ or (iii) $CuO \rightarrow Cu$. The presence of Cu is also confirmed by nuclear magnetic resonance spectroscopy in the 7 h- and 18 h-milled samples (not shown here). When the $CeO_2$ and CuO powder phases are forced to contact at the bounding interphase interface during milling the cations of $Ce^{4+}$ (or $Ce^{3+}$ for non-stoichiometric sites especially near the surface) could be interchanged with $Cu^{2+}$ cations through the vacancy mechanism. The $CeO_2$ powder, in particular the surface volume, exhibits non-stoichiometric compositions with various defects [42], which can be formed upon the introduction of metal cations with higher or lower valences into $CeO_2$. It is also known that the mechanical milling of powders induces the fracture and deformation through high-energy collisions among balls, vial surface and particles [43], leading to a modification of the crystal structure as well as large lattice strain. In this study, the Cu cations can be introduced in the lattice of $CeO_2$, replacing the Ce cations as well as producing the extra oxygen vacancies in the $CeO_2$ lattice.

Figure 2 shows XRD patterns of 18 h-milled samples with various $CeO_2$–CuO content ratios. For pure CuO (figure 2a), no phase change of CuO is observed after 18 h milling. But, with the 80 mol% CuO composite (figure 2b), the coexistence of both Cu and $Cu_2O$ phases is observed replacing the CuO, besides the existence of the cubic $CeO_2$. With 70 mol% CuO (figure 2c), the observable Cu-related phase after 18 h milling is an fcc Cu phase only. By further reducing the CuO content (figure 2d–f) for each after 18 h milling, the emerged intensity of the Cu phase is gradually lowered, without a major change in the $CeO_2$ reflections.

The lattice parameter and the estimated crystallite size of the cubic $CeO_2$ phase in the $(CuO)_{0.5}(CeO_2)_{0.5}$ composite are shown in figure 3, as a function of milling time. The crystallite size is rapidly decreased during an early stage of the milling periods, less than approximately 5 h, followed by a gradual decrease up to 30 h milling. But the lattice parameter is only gradually increased during an early stage of milling, less than approximately 5 h, followed by a rather rapid increase with the

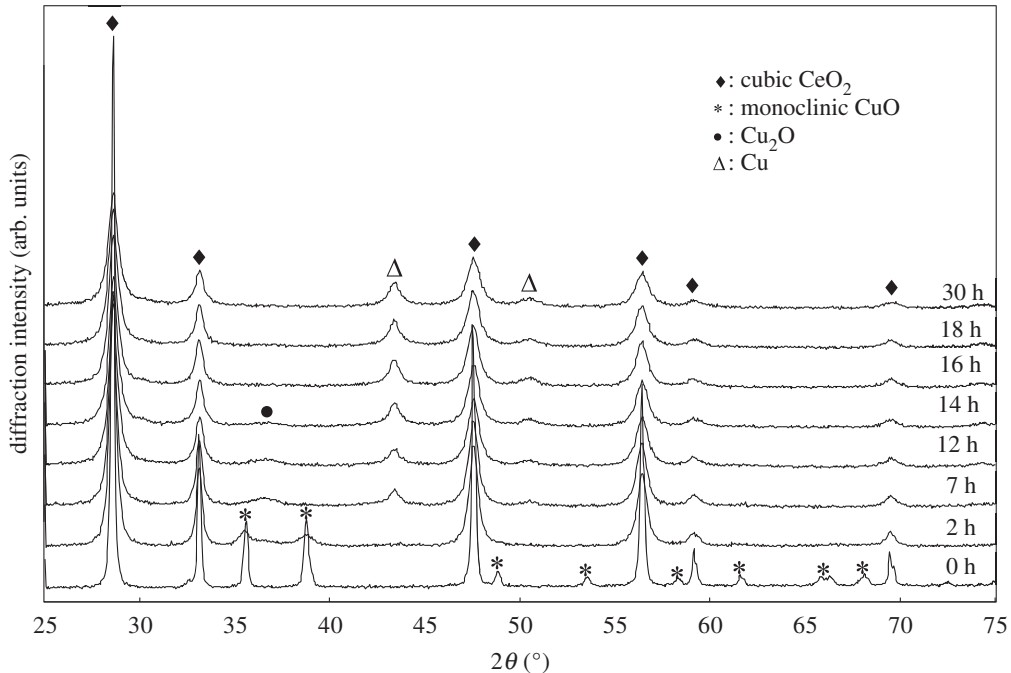

**Figure 1.** XRD patterns of $(CuO)_{0.5}(CeO_2)_{0.5}$ powder (CuO: monoclinic, CeO$_2$: cubic) with milling time.

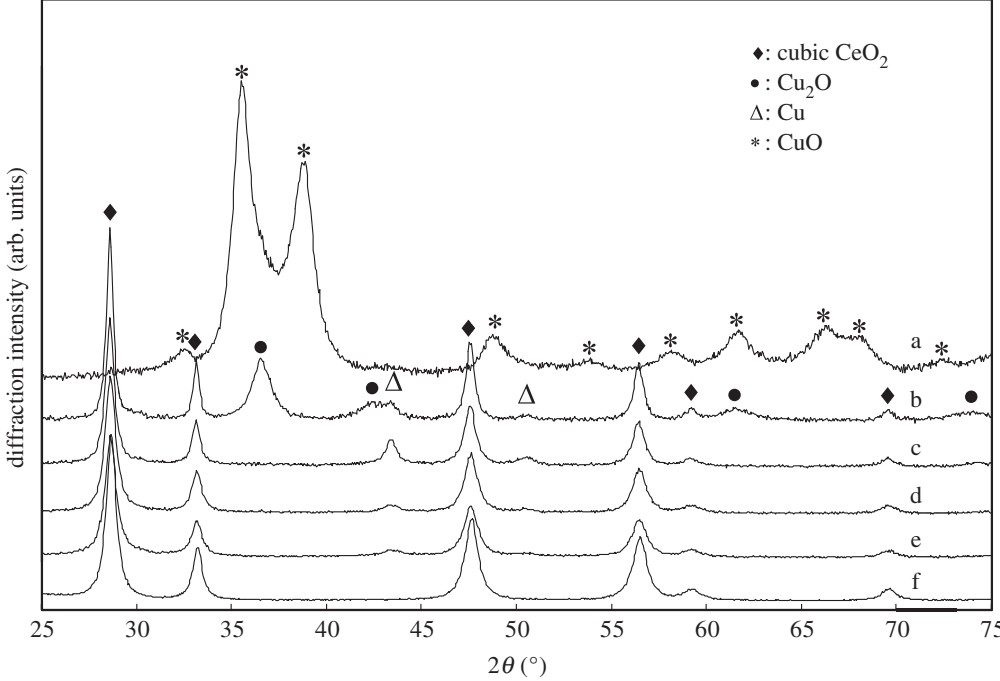

**Figure 2.** XRD patterns $(CuO)_x(CeO_2)_{1-x}$ powder after 18 h milling ($x=$: (a) 1, (b) 0.8, (c) 0.7, (d) 0.5, (e) 0.2, (f) 0).

increase of the milling time up to approximately 18 h, and then the variation becomes small up to 30 h milling. This indicates that although the change of the crystallite size is not directly related to the lattice parameter variation, the latter is activated only after the former event. This is consistent with the idea that the appearance of Cu and Cu$_2$O phases is clearly observed when the milling duration is longer than 7 h, as shown in figure 1, because atomic-order mixing of powders that is prerequisite for the phase change is generally expected, particularly in the vicinity of the interphase interface, only after effective repeated folding of particles during milling, the so-called kneading effect, producing fine layered structures inside powder. It is also suggested that a steady state of the structural variation of CeO$_2$ is attained after approximately 18 h milling. The decrease of crystallite size and the increase of lattice parameter of CeO$_2$ with milling time are also reported [44–48].

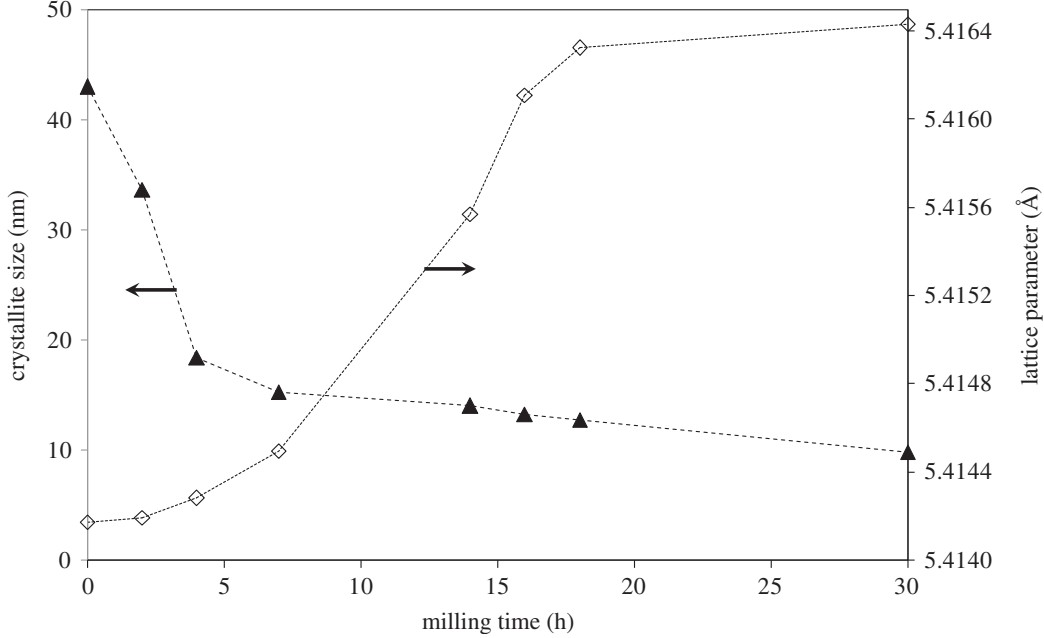

**Figure 3.** Crystallite size and lattice parameter of $CeO_2$ in $(CuO)_{0.5}(CeO_2)_{0.5}$ composite powder as a function of milling time.

**Table 1.** Lattice parameter, crystallite size, of cubic $CeO_2$ and BET surface area for various $(CuO)_x(CeO_2)_{1-x}$ samples after 18 h milling.

| sample | lattice parametter ($A^0$) | crystallite size (nm) d[220] | $S_A$ (BET) $m^2g^{-1}$ |
|---|---|---|---|
| $(CuO)_{0.8}(CeO_2)_{0.2}$ | 5.4168(6) | 11.3 | 17.5 – 19.7 |
| $(CuO)_{0.5}(CeO_2)_{0.5}$ | 5.4163(2) | 12.8 | 18.6 – 21.9 |
| $(CuO)_{0.8}(CeO_2)_{0.2}$ | 5.4160(0) | 13.6 | 19.4 – 22.3 |
| $(CuO)_{0.2}(CeO_2)_{0.8}$ | 5.4158(9) | 14.4 | 21.1 – 23.7 |
| $CeO_2$ | 5.4152(4) | 15.6 | 23.3 – 25.6 |

The lattice parameters and crystallite sizes of the cubic $CeO_2$ phase and the specific surface areas (BET) of 18 h-milled samples with various CuO contents are listed in table 1. With an increase of mol% CuO contents, the lattice parameter of $CeO_2$ is increased, the average crystallite size is decreased and the BET surface area of $CuO–CeO_2$ powder composite is decreased. Because the smaller crystallite size may cause an elongation of the cubic lattice in the nano-sized $CeO_2$ phase [47], which is ascribed to the lattice strain from the formation of the $Ce^{3+}$ cations and the corresponding oxygen vacancies, the observed tendency is consistent. The slight decrease of the BET area with the CuO ratio is probably due to formation of the soft Cu phase during 18 h milling (figure 2), which may cause agglomeration of powders.

## 3.2. Morphological features

The morphological features and the compositional homogeneity are studied by SEM and EDX. Similar morphology is observed for all the milled samples, where aggregations of packed particles from a few hundred nanometres to micrometres in size exist, as shown in figure 4. Some reports show that the CuO is incorporated and evenly distributed into the $CeO_2$ structure [49,50]. As shown in figure 5, with the use of each beam size adjusted as 10 μm, the areal average concentrations of each 18 h-milled sample with a different overall composition (20, 50, 70 or 80 mol% CuO) exhibit high homogeneity in terms of Ce and Cu contents, judging from the EDX analyses at 20 randomly selected squares. The magnification of SEM is fixed at 220 times. This strongly suggests that the dimensional size of the Cu or $Cu_2O$ phase, as observed in figure 2, is much smaller than 10 μm. Contamination from the milled media ($ZrO_2$ balls

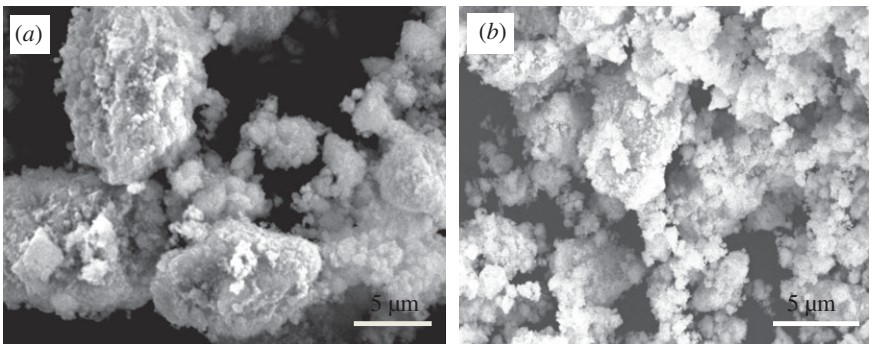

**Figure 4.** SEM micrograph of 18 h-milled powders; (a) $(CuO)_{0.8}(CeO_2)_{0.2}$ and (b) $(CuO)_{0.2}(CeO_2)_{0.8}$.

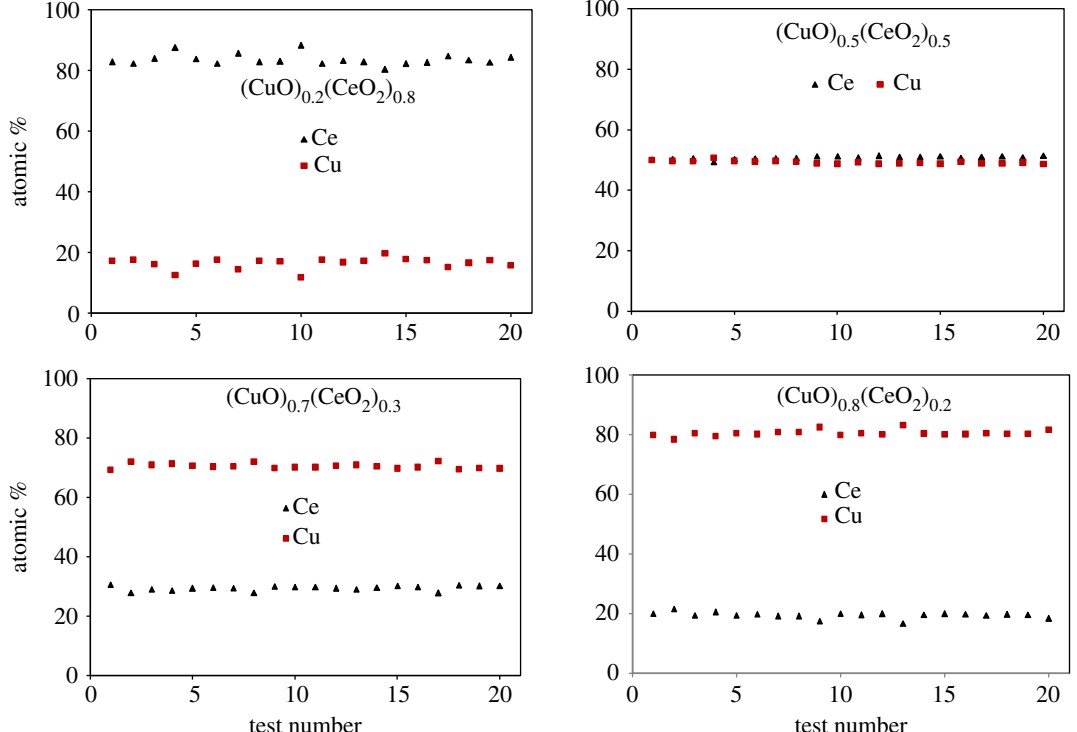

**Figure 5.** EDX analysis on various $(CuO)_x(CeO_2)_{1-x}$ samples after 18 h milling; 15 kV.

and a steel vial) is also examined by EDX, and the maximal contamination level is less than 1 wt% for Fe while no contamination is detectable for Cr, $ZrO_2$, and so on, after 18 h milling.

## 3.3. Reduction behaviour by TPR

The $H_2$-TPR profiles of the $(CuO)_{0.5}(CeO_2)_{0.5}$ samples milled for 0, 4, 7, 14 and 18 h are shown in figure 6. For powders without milling (0 h milling), i.e. by mixing the oxide powders, the reduction of $(CuO)_{0.5}(CeO_2)_{0.5}$ is characterized by rather combined three peaks ($\alpha$, $\beta$, and $\gamma$) in the range of 220–420°C and one broad $\phi$ peak in the range of 700–880°C. After 4 h milling, the former peaks ($\alpha$, $\beta$ and $\gamma$) merge into one large broad peak cantered around 300°C–320°C with some skewed symmetry. The position and the intensity of the peak are not largely changed with further milling, but the skewness is somewhat increased, with the peak top shifting to the higher temperatures with milling time up to 18 h (figure 6). The progression of the milling process should produce a large amount of bounding interface/interphase between CuO and $CeO_2$, and some Cu atoms would even be penetrated into $CeO_2$, producing $Ce_{1-x}Cu_xO_2$ solid solutions near the interface vicinity. These would facilitate the mobility of oxygen atoms for both phases, resulting in promoted valence variation of cations and leading to the observed large areas of $H_2$ consumption peak. As for the latter peak, the broadness of the peak is slightly reduced with milling time, whereas the peak top is around 800°C.

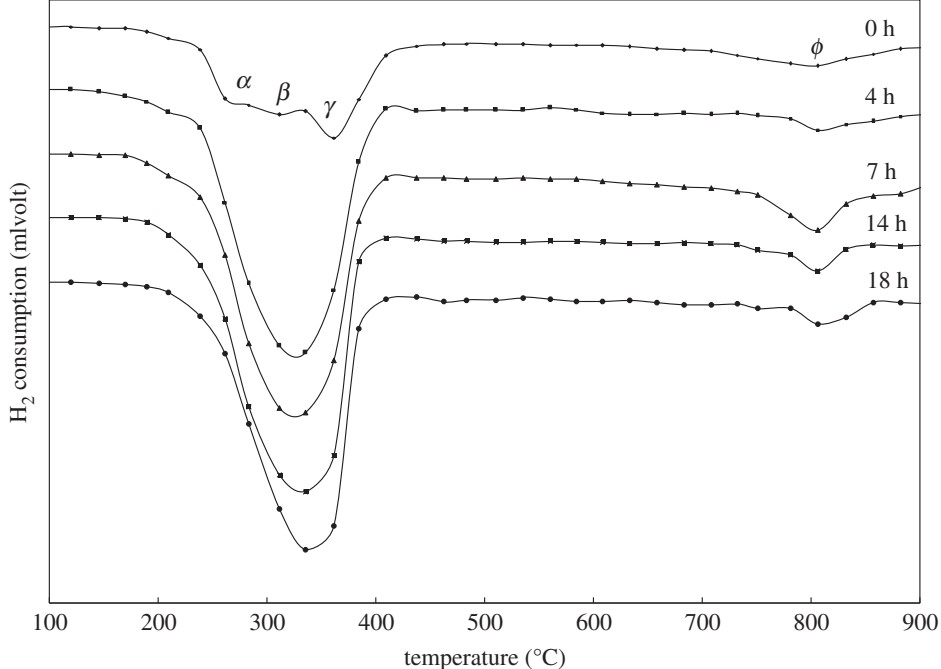

**Figure 6.** TPR of $(CuO)_{0.5}(CeO_2)_{0.5}$ composite powder with milling time.

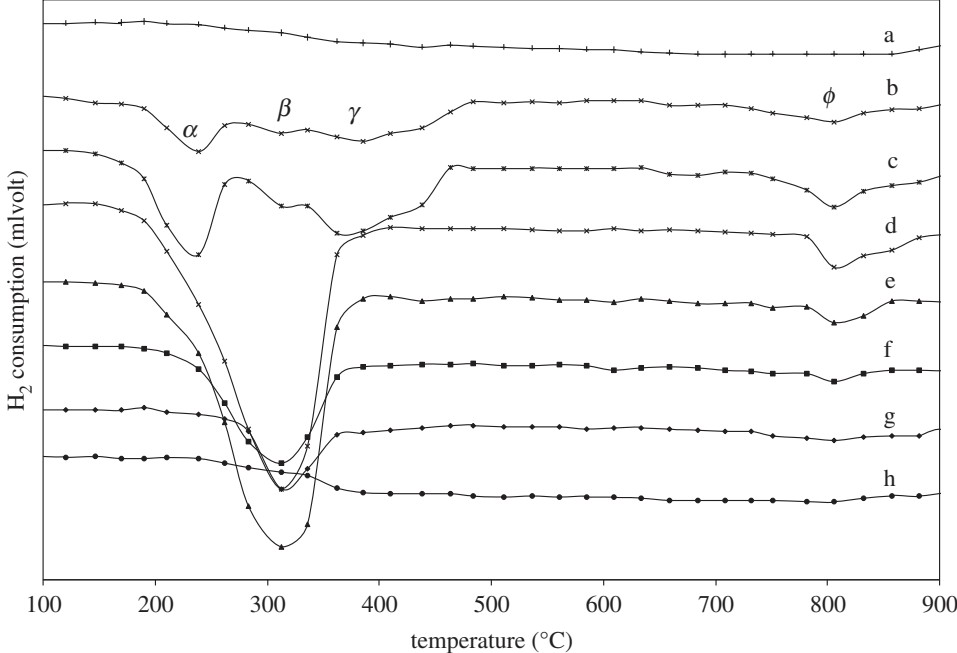

**Figure 7.** TPR of (a) $CeO_2$, (b) CuO powder for 0 h milling and $(CuO)_x(CeO_2)_{1-x}$ powder after 18 h milling ($x=$ (c) 1, (d) 0.8, (e) 0.5, (f) 0.3, (g) 0.2, (h) 0).

Figure 7 shows the $H_2$-TPR profiles of various $(CuO)_x(CeO_2)_{1-x}$ powders ($x = 1, 0.8, 0.5, 0.3, 0.2, 0$) after 18 h milling, compared with pure $CeO_2$ (a) and CuO (b) powders without milling (0 h milling). For pure CuO ($x = 1$), the reduction is characterized by rather combined three peaks ($\alpha$, $\beta$, and $\gamma$) in the range of 200–470°C and one broad $\phi$ peak in the range of 750–850°C (figure 7b,c), regardless of milling or without milling, strongly indicating that the powder morphology and the milling-induced defects are not the major factors influencing the reduction behaviour. In addition, the range of reduction temperatures is wider compared with that of the $(CuO)_{0.5}(CeO_2)_{0.5}$ composite (figure 6: 0 h), indicating multiple reduction steps are involved with possible inhomogeneous reactions. But, only two apparent peaks are observed for lower CuO contents (figure 7d–g), and the intensity of both peaks is reduced when CuO content is decreased, demonstrating the major contribution of CuO to the $H_2$

consumption. For pure $CeO_2$ (figure 7a,h), there is no distinguished $H_2$-TPR peak in the temperature range observed, while the consumption of $H_2$ steadily exists above 350°C up to approximately 900°C for reducing $CeO_2$ to $Ce_2O_3$, where the reduction first occurs near the surface defects [51,52] followed by the formation of intermediate CeO2-x and complete transformation to $Ce_2O_3$.

It is reported by Fierro et al. [53] that the TPR characteristics could be affected by mass transfer limitations and experimental operating variables such as the initial amount of reducible species, the initial $H_2$ concentration, the total gas flow rate, the heating rate, and the activation energy of the reaction. They claim that desorption of H2 attached to the reduced Cu metal surface could exhibit the apparent double-peak behaviour, which may be affected by water vapour produced by the reduction process, and that the $H_2$-TPR profile of CuO depends on the particle size and surface area, where the peak top is higher by 288°C for particle sizes of 425 approximately 850 microns compared with less than 100 microns [53]. A similar tendency but with a much larger peak shift (over 373°C) is also reported by Luo et al. [49], who claim that the hydrogen spillover effect is the reason for the difference between CO-TPR and $H_2$-TPR.

Kim et al. [54] report that there is an incubation period prior to reduction, which is longer at lower temperatures. The tendency is in agreement with the general theory of nucleation and growth, where the number of newly formed nuclei is copious at lower temperatures, but because the growth is very much limited at lower temperatures, the phase existence is not easily detected by x-ray diffractometry. They claim that CuO reduction is generally easier than $Cu_2O$ reduction with $H_2$-TPR, with the apparent activation energy for $Cu_2O$ close to twice that of CuO, but when the $H_2$ flow rate is not high enough for avoiding the rate-limiting of the reduction process, a sequential reduction process such as $CuO \rightarrow (Cu_4O_3 \rightarrow) Cu_2O \rightarrow Cu$ may happen. Our experimental condition is: 15°C min$^{-1}$, 50 mg and 25 cc min$^{-1}$ of 5%$H_2$ flow. This is close to the condition for the appearance of the sequential process due to the 'lean $H_2$' flow. The incubation period and the sequential/simultaneous reduction process are also reported by Kim et al. [54], where, with the lean $H_2$ condition, the CuO, $Cu_2O$ and Cu are simultaneously observed after a certain incubation period according to the time-resolved XRD. This is consistent with our thought experiments (not shown here), where the 3 phases are simultaneously observed when the H2-TPR is stopped at 280°C and kept for 15 min, while only CuO is observed when the sample is immediately quenched from 280°C to RT.

Consequently, the $\alpha$ peak for pure CuO reduction, as shown in figure 7, should be mainly attributed to direct reduction of CuO particles into Cu in the surface vicinity, and some simultaneous reduction to $Cu_2O$ (or $Cu_4O_3$) should be involved with our experimental operating variables, somewhat contributing to the $\alpha$ peak. Because the rate of nucleation and growth of each reduced phase is different, which involves the shape of nuclei and the nucleation sites besides the spillover effect of $H_2$ on the metallic Cu, one rate-determining step would gradually dominate the sequential CuO reduction to $Cu_2O$ and Cu, exhibiting the broad β peak for the slow reduction process, which is related to the larger activation energy for $Cu_2O$ formation compared with direct Cu formation. Before the end of the sequential reaction, some $H_2$ molecules would be dissociated from H atoms on newly formed Cu on the powder surface, penetrating into the $Cu(/Cu_2O)/CuO$ particles, and at certain temperature(s), depending on the surrounding condition, the remaining CuO would be reduced to Cu either directly or sequentially through $Cu_2O$ (or $Cu_4O_3$), exhibiting the γ peak. The extended TPR peak profile as high as 470°C or even 480°C strongly indicates the difficulty of reduction due to longer the diffusion path for H atoms or $H_2$ molecules as well as the escape of produced $H_2O$ molecules. It is acknowledged that the larger the CuO particles, the thicker the reduced Cu phase; this covers the particle and shows the bulk-like behaviour of CuO reduction. Thus, the $\alpha$ and $\beta$ peaks are probably caused by surface-vicinity reduction with separate locations, whereas the γ peak would be due to reduction inside the particles. At the high temperature $\phi$ peak, it may cause the Cu surface to interact with the $SiO_2$ quartz reactor [55,56].

The apparent increase of the $H_2$-TPR starting temperature for the $\alpha$ peak, as shown in figure 6, compared with pure CuO in figure 7b,c is probably due to variation on the mass transfer limitations affected by experimental operating variables such as the initial amount of reducible species and effective total gas flow rate through mixing $CeO_2$ particles with CuO particles. But, the end temperature for the γ peak, as shown in figure 6, is much decreased, compared with pure CuO shown in figure 7b,c. This must be the mixing effect of $CeO_2$ particles with CuO particles, in that the contact point or the interphase interface plays an important role in the nucleation and growth of the reduced phase such as Cu, or even $Cu_2O$. Owing to the valence change of cations requiring high electron mobility as well as the higher oxygen mobility near the contact point, the nucleation event is much more active in the vicinity, and with the fast surface diffusion of Cu atoms the nucleation is

immediately followed by growth or accumulation of the new phase, always leaving some area of fresh CuO surface contacting with $H_2/Ar$ gas. In the reduction process, the diffusion paths for H atoms or $H_2$ molecules as well as the escape path of the produced $H_2O$ molecules are established, and the reduction is continued until all the CuO is consumed.

There are many reports on the $CuO-CeO_2$ powder systems and the reduction study. Many researchers report that ceria in the $CuO-CeO_2$ powder promotes surface species reduction of highly dispersed copper oxide [57–64] and the CuO particles with smaller sizes are easier to reduce [65]. According to Liu *et al.* [58], the reduction peak at lower temperature is attributed to copper oxide clusters strongly interacting with ceria, whereas the peak at higher temperature is attributed to larger CuO particles, not associated with ceria. A similar tendency in the $H_2$-TPR profile is also reported by Luo *et al.* [60] and Xiaoyuan *et al.* [61]. Furthermore, Tang *et al.* [66] report that the lower temperature peak is assigned to reduction of amorphous CuO clusters closely interacting with the $CeO_2$, while Luo *et al.* [48] claim that with increasing CuO content three peaks appear, with the lowest temperature deriving from reduction of finely dispersed CuO, the intermediate temperature associated with reduction of $Cu^{2+}$ ions in $Cu_xCe_{1-x}O_{2-\delta}$ solid solutions, and the additional highest one due to reduction of bulk CuO.

As observed in figure 6, with increasing milling time on $(CuO)_{0.5}(CeO_2)_{0.5}$ composite samples, the skewness of $H_2$-TPR profile is somewhat increased with the peak top shifting to the higher temperatures. Also, in figure 7, as the CuO content is decreased in various 18 h-milled $(CuO)_x(CeO_2)_{1-x}$ powders, the $H_2$ absorption at lower temperatures involved with the $\alpha$ peak is gradually diminished, with the peak tops of the main $H_2$-TPR profiles around 310–330°C, which is similar to that shown in figure 6. Wang *et al.* claim that [49] Cu atoms embedded in ceria have an oxidation state higher than those of the cations in $Cu_2O$ or CuO, where (i) Cu in the doped sample is only partially reduced to metallic Cu, especially at low temperatures as $CuO \rightarrow Cu_2O \rightarrow Cu$, because Cu embedded in ceria is difficult to reduce, (ii) The lattice of the $Ce_{1-x}Cu_xO_2$ systems (fluorite-type) is highly distorted with local order defects and multiple cation-oxygen distances, and (iii) Doping $CeO_2$ with Cu introduces large strain and O vacancies.

This is consistent with our results in that, at the early stage of milling such as before 7 h milling, the ultrafine mixture of $CuO/CeO_2$ is more effective for $H_2$-TPR profile, which corresponds to the rapid crystallite size reduction before 4 h milling, as shown in figure 3. Here, the large area of $CuO/CeO_2$ interphase interface is extremely efficient for producing the path for $H_2$ diffusion and $H_2O$ escape, besides the effective valence change of cations, where the x-ray diffractometry in figure 1 identifies only CuO and $CeO_2$ phases. But, the longer the milling time, the more intense the formation of $Ce_{1-x}Cu_xO_2$ solid solution systems, where the higher peak temperature for the $H_2$-TPR profile should be observed due to Cu embedded in ceria as milling proceeds. The results are also consistent with our x-ray results shown in figure 1, where atomic-order mixing is achieved at least near the interface vicinity of $CuO/CeO_2$, producing Cu and $Cu_2O$ after 7 h milling, and that the clear increase in the lattice parameter is observed (figure 3) after approximately 7 h milling.

It is thus quite reasonable to consider that, at the early stage of milling, the major $H_2$-TPR peaks caused by milling of $CuO/CeO_2$ are attributed to reduction of $Cu^{2+}$ ions in the interfacial vicinity of $CeO_2$, which exist in the ultrafine mixture of $CuO/CeO_2$ or some $Cu_2O/CeO_2$. The CuO particles are highly dispersed, producing many active sites for $Cu^{2+}$ reduction on the interphase interface with $CeO_2$. With the increase of milling time, some Cu atoms are even incorporated into $CeO_2$, causing formation of $Ce_{1-x}Cu_xO_2$ solid solutions. The large intensity and the broad profile of the $H_2$-TPR peaks would be observed due to the dual effect.

Figure 8 shows the repeated $H_2$-TPR cycles of milled $(CuO)_{0.5}(CeO_2)_{0.5}$ composites (50 mol% CuO milled for 0, 4, 7, 14 and 18 h) in figure 6, each after the first TPR. The peak intensity for the 0 h milled sample is largely decreased, when compared with the first run, with high temperature peak(s) remaining, suggesting agglomeration/recrystallization of CuO particles that produce bulk-like CuO after heating to 900°C. When increasing the milling time, the $CeO_2$ particles are finely distributed into the CuO particles, or some Ce–Cu–O solid solutions are produced, which prevents agglomeration/ recrystallization of CuO, leading to the slow decrease observed in the peak intensity. The temperatures of the reduction start as well as the peak top of the $H_2$ consumption are lowered compared with the first run. On considering that the pure CuO is reduced in the wide range of temperatures (200–470°C, figure 7*b,c*) regardless of milling or without milling under this particular experimental condition, some CuO is heterogeneously reduced (lean $H_2$-TPR) at low temperatures (200–250°C) to produce Cu and $Cu_xO$ simultaneously. But, because the $(CuO)_{0.5}(CeO_2)_{0.5}$ composite powders are finely mixed with milling, even after the $H_2$-TPR to 900°C, the elemental mixture is maintained, and the CuO reduction is activated by surface contact with $CeO_2$.

...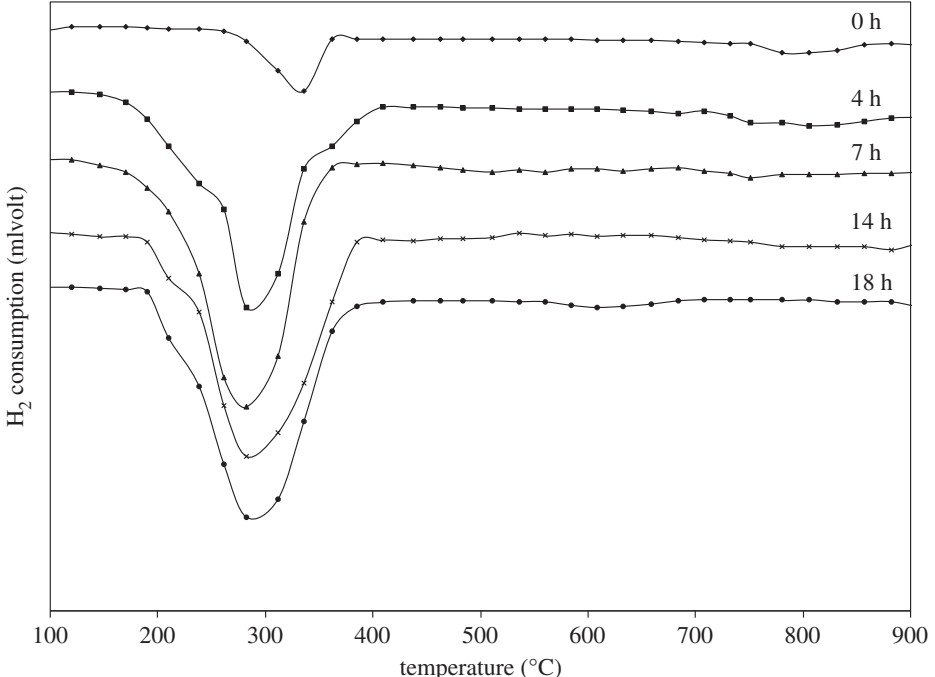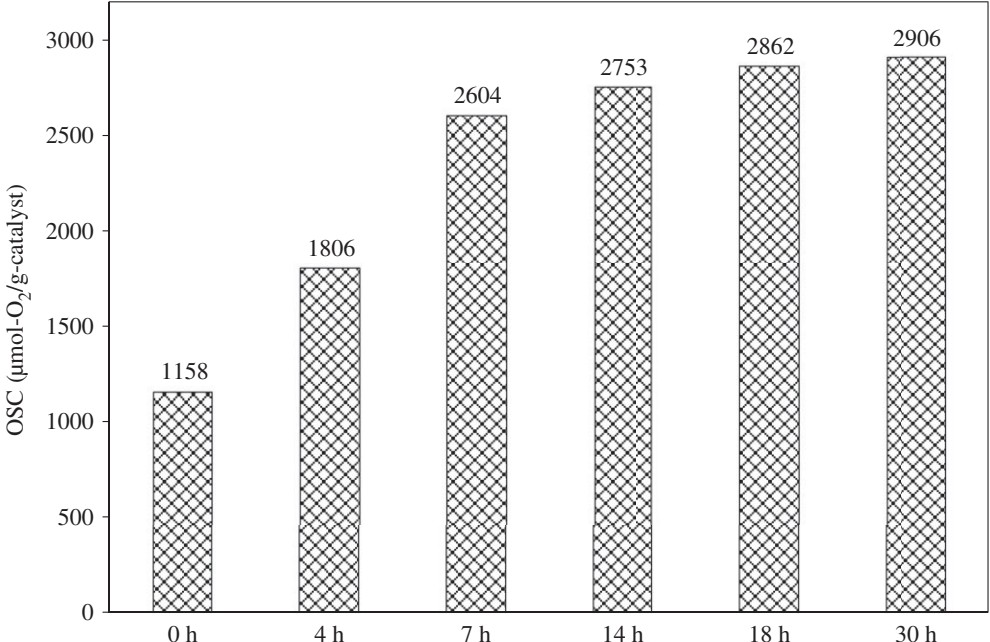

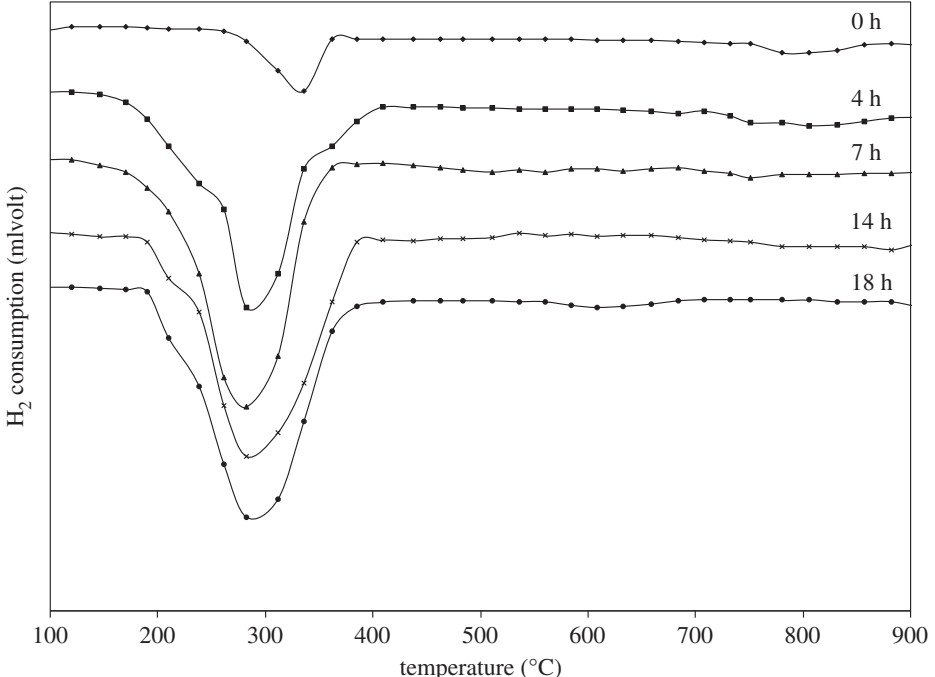

**Figure 8.** TPR of $(CuO)_{0.5}(CeO_2)_{0.5}$ powder with milling time after the first TPR cycle.

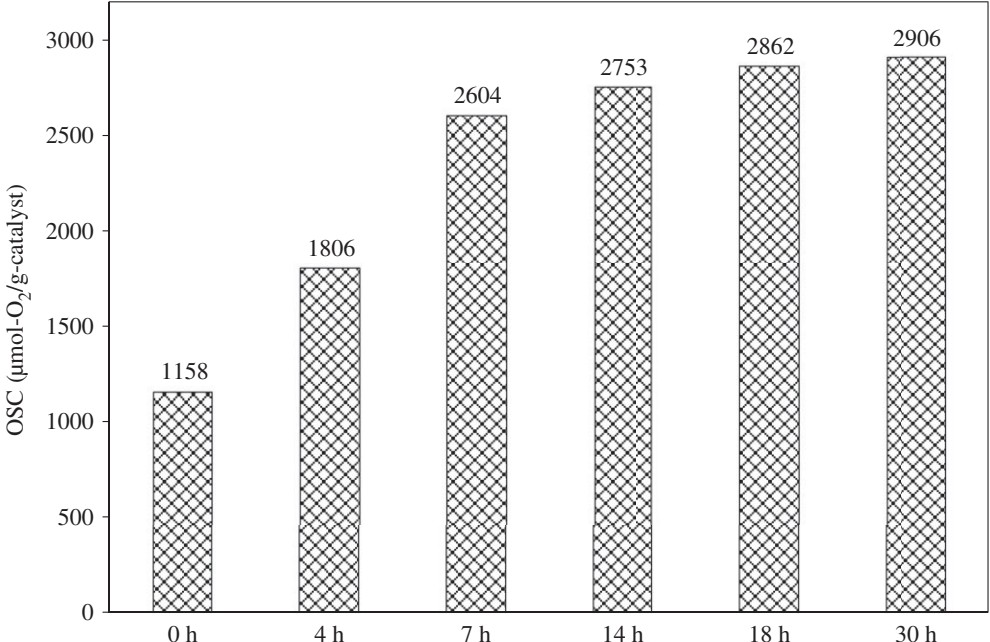

**Figure 9.** Total OSC at 300°C of $(CuO)_{0.5}(CeO_2)_{0.5}$ powder with milling time.

Figure 9 shows the total OSC at 300°C of various milled samples (50 mol% of CuO) after 0, 4, 7, 14, 18 and 30 h milling. The total OSC is increased with milling time, with the rapid increase from 0 to 7 h milling followed by the gradual increase from 7 h to 30 h. This is consistent with figure 1 (x-ray results) and figure 3 in that the rapid atomic-order mixing is effective during the early stage milling of 7 h, evidenced by both solid state reactions and crystallite size variations. It is also noted that, because the lattice parameters are largely varied after 7 h milling, the $CuO/CeO_2$ interphase interface creation and the corresponding atomic contact would be more important for the effective OSC than formation of the Ce–Cu–O solid solution, although both contribute to the OSC promotion. That is, the activated valence change of $Cu^{2+}/Cu^{+}/Cu$ assisted by neighbouring Ce-O bonds is extremely important, and the oxygen transport path is also critical for the OSC. The enhanced oxygen storage/transport would

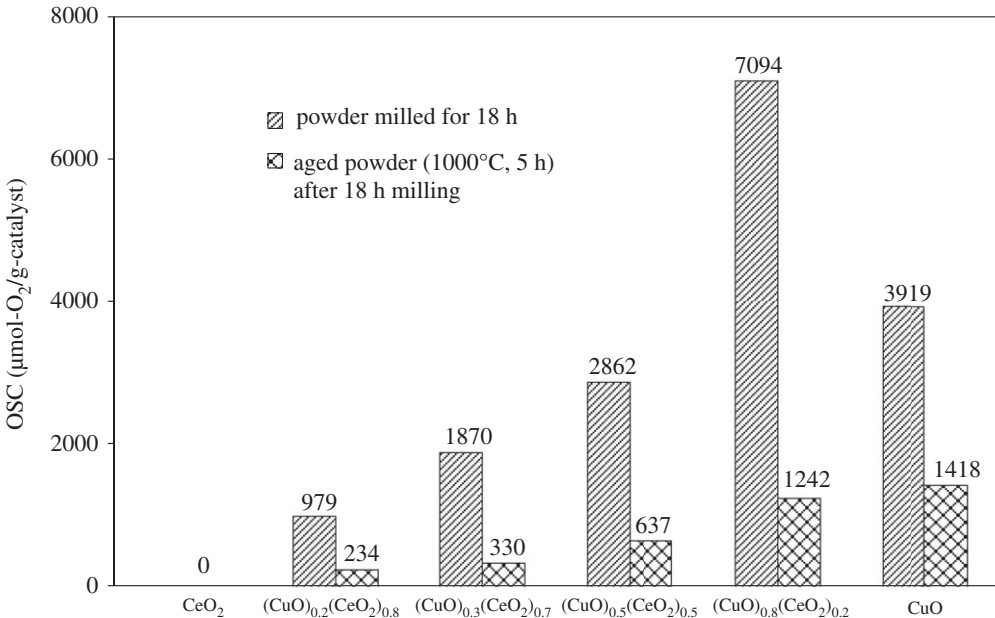

**Figure 10.** Total OSC at 300°C of milled samples for 18 h and after ageing at 1000°C for 5 h.

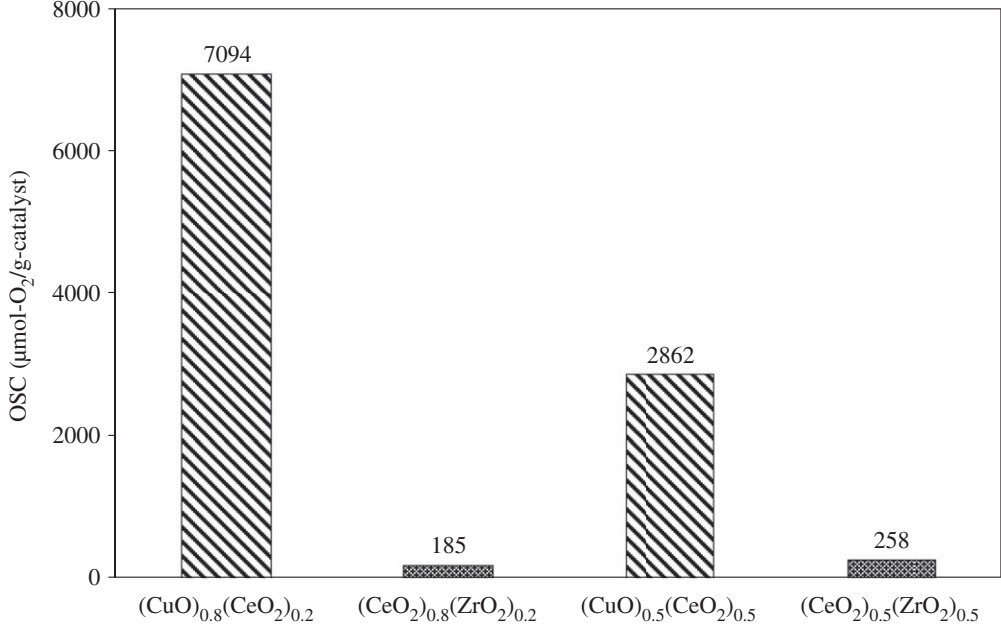

**Figure 11.** Comparison of total OSC at 300°C of CuO−CeO$_2$ and CeO$_2$−ZrO$_2$-milled powder.

thus be realized through easy valence change of $Cu^{2+}/Cu^{+}/Cu$ neighboured by Ce-O bonds and the large surface area with the large number of the available oxygen vacancies.

The total OSC at 300°C for the samples milled for 18 h is increased, as shown in figure 10, from 0 to 7094 μmol-O$_2$/g-catalyst with increased CuO content from 0 to 80 mol%, but is decreased to 3919 μmol-O$_2$/g-catalyst for pure CuO (BET surface area of 15–16 m$^2$ g$^{-1}$). After ambient ageing at 1000°C for 5 h, however, the total OSCs at 300°C are significantly reduced (figure 10), probably ascribed to the agglomeration/recrystallization of fine particles.

As shown in figure 11, the total OSCs of the CeO$_2$-20 at %ZrO$_2$ and the CeO$_2$-50 at %ZrO$_2$ at 300°C are 185 and 258 μmol-O$_2$/g-catalysts, respectively, while those of the CuO−CeO$_2$ catalyst system prepared with the same condition are at least one order greater. It clearly shows that the mechanically driven CuO/CeO$_2$ system exhibits the high OSC property, and this should contribute to improving the performance of TWCs at lower temperatures.

# 4. Conclusion

Mechanical milling was applied to the CuO–CeO₂ powder system to produce mixed-oxide catalysts. The milled sample was characterized by the use of XRD, SEM, TG-DTA, GC-TCD and BET analyses. Milling of powder mixtures of CuO and CeO₂ showed the reduction of CuO when milling was processed in air. The crystallite size of ceria was rapidly decreased at the early stage of milling, followed by an increase of the lattice parameter, indicating formation of $Ce_{1-x}Cu_xO_2$ solid solutions after the rapid crystallite size reduction. The redox property of milled CuO–CeO₂ samples was investigated by $H_2$-TPR. Three reduction peaks were observed for 0 h milling and only one broad peak for various milling times, where the valence change of Cu ions enhanced the redox activity. The higher OSC for the CuO–CeO₂ system was observed with increased milling time. The total OSC of the CuO–CeO₂ catalyst was much higher than that of the CeO₂–ZrO₂ traditional catalyst system at low temperature.

Data accessibility. The datasets supporting this article have been uploaded as part of the electronic supplementary material.

Authors' contributions. Nguyen The Luong carried out the catalyst preparation, experiment tests and drafted the manuscript. Hideyuki Okumura, Eiji Yamasue and Keiichi N. Ishihara participated in the design of the study and helped to modify the manuscript. All authors approved the final version of the manuscript.

Competing interests. The authors declare no competing interests.

Funding. This work was supported by Global Center of Excellence (GCOE) Program and the Monbukagakusho Scholarship, Japan.

Acknowledgments. The authors thank Prof. K. Yoshimura and Prof. C. Michioka for their help on NMR measurements.

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
