## [Reviewer comments · Royal Society Open Science]

Review History

RSOS-181861.R0 (Original submission)

Review form: Reviewer 1 (Trung Tran)

Is the manuscript scientifically sound in its present form?

Yes

Are the interpretations and conclusions justified by the results?

Yes

Is the language acceptable?

Yes

Is it clear how to access all supporting data?

Yes

Do you have any ethical concerns with this paper?

No

Have you any concerns about statistical analyses in this paper?

No

Recommendation?

Accept with minor revision (please list in comments)

Comments to the Author(s)

I would like to recommend this article to publish in the Journal with minor revisions as attached file (Appendix A).

Review form: Reviewer 2

Is the manuscript scientifically sound in its present form?

Yes

Are the interpretations and conclusions justified by the results?

Yes

Is the language acceptable?

Yes

Is it clear how to access all supporting data?

Yes

Do you have any ethical concerns with this paper?

No

Have you any concerns about statistical analyses in this paper?

No

Recommendation?

Major revision is needed (please make suggestions in comments)

Comments to the Author(s)

1. In line 56: the authors stated that "Although not detected, some CuO phase may exist as nanostructured or amorphous state." There is no experimental indication of this statement. The authors should confirm this statement experimentally.
2. There is no stability work for CeO₂-CuO system in the manuscript. It is known that although the presence of CuO besides CeO₂ enhances the OSC values compared to CeO₂-ZrO₂ or CeO₂-TiO₂, the thermal stability of the CuO-containing samples is quite low. The authors should discuss this in their manuscript.
3. The total OSC of CeO₂-CuO system is not directly related to the structural oxygen defects present in CeO₂ system. It is also related to the high reducibility of the CuO phase. If the authors calculated theoretical the OSC values, they would also realize this effect.
4. The authors did not use XPS and Raman Spec in the study to show the effect of mechanical milling on the density of structural oxygen defects in metal oxide particle system. The results obtained from the mentioned experimental methods will shed light on the enhanced OSC property and improve the author's results.
5. Some of the beneficial references are missing in the manuscript. The following papers can be

used to improve the statements and address some of the concerns listed above. Therefore, they should be cited in the manuscript.

Uzunoglu, A., Zhang, H., Andreescu, S., Stanciu, A. L., CeO₂-MO_x (M: Zr, Ti, Cu) mixed metal oxides with enhanced oxygen storage capacity, *Journal of Materials Science* (2015) 50: 3750-3762, DOI 10.1007/s10853-015-8939-7.

Uzunoglu, A., Kose, DA., Stanciu, LA., Synthesis of CeO₂-based core/shell nanoparticles with high oxygen storage capacity; *International Nano Letters*, 2017, 7:187-193.

Decision letter (RSOS-181861.R0)

27-Nov-2018

Dear Dr Luong:

Title: Structure and catalytic behavior of CuO-CeO₂ prepared by High-Energy Ball Milling
Manuscript ID: RSOS-181861

The editor assigned to your manuscript has now received comments from reviewers. We would like you to revise your paper in accordance with the referee and Subject Editor suggestions which can be found below (not including confidential reports to the Editor). Please note this decision does not guarantee eventual acceptance.

Please submit your revised paper before 20-Dec-2018. Please note that the revision deadline will expire at 00.00am on this date. If we do not hear from you within this time then it will be assumed that the paper has been withdrawn. In exceptional circumstances, extensions may be possible if agreed with the Editorial Office in advance. We do not allow multiple rounds of revision so we urge you to make every effort to fully address all of the comments at this stage. If deemed necessary by the Editors, your manuscript will be sent back to one or more of the original reviewers for assessment. If the original reviewers are not available we may invite new reviewers.

Please also include the following statements alongside the other end statements. As we cannot publish your manuscript without these end statements included, if you feel that a given heading is not relevant to your paper, please nevertheless include the heading and explicitly state that it is not relevant to your work.

• Ethics statement

Please clarify whether you received ethical approval from a local ethics committee to carry out your study. If so please include details of this, including the name of the committee that gave consent in a Research Ethics section after your main text. Please also clarify whether you received informed consent for the participants to participate in the study and state this in your Research Ethics section.

OR

Please clarify whether you obtained the necessary licences and approvals from your institutional animal ethics committee before conducting your research. Please provide details of these licences and approvals in an Animal Ethics section after your main text.

OR

Please clarify whether you obtained the appropriate permissions and licences to conduct the fieldwork detailed in your study. Please provide details of these in your methods section.

RSC Associate Editor:
Comments to the Author:
(There are no comments.)

RSC Subject Editor:
Comments to the Author:
(There are no comments.)

Reviewers' Comments to Author:
Reviewer: 1

Comments to the Author(s)
I would like to recommend this article to publish in the Journal with minor revisions as attached file

Reviewer: 2

Comments to the Author(s)

1. In line 56: the authors stated that “Although not detected, some CuO phase may exist as nanostructured or amorphous state.” There is no experimental indication of this statement. The authors should confirm this statement experimentally.
2. There is no stability work for CeO₂-CuO system in the manuscript. It is known that although the presence of CuO besides CeO₂ enhances the OSC values compared to CeO₂-ZrO₂ or CeO₂-TiO₂, the thermal stability of the CuO-containing samples is quite low. The authors should discuss this in their manuscript.
3. The total OSC of CeO₂-CuO system is not directly related to the structural oxygen defects present in CeO₂ system. It is also related to the high reducibility of the CuO phase. If the authors calculated theoretical the OSC values, they would also realize this effect.
4. The authors did not use XPS and Raman Spec in the study to show the effect of mechanical milling on the density of structural oxygen defects in metal oxide particle system. The results obtained from the mentioned experimental methods will shed light on the enhanced OSC property and improve the author’s results.
5. Some of the beneficial references are missing in the manuscript. The following papers can be used to improve the statements and address some of the concerns listed above. Therefore, they should be cited in the manuscript.

Uzunoglu, A., Zhang, H., Andreescu, S., Stanciu, A. L., CeO₂-MO_x (M: Zr, Ti, Cu) mixed metal oxides with enhanced oxygen storage capacity, *Journal of Materials Science* (2015) 50: 3750-3762, DOI 10.1007/s10853-015-8939-7.

Uzunoglu, A., Kose, DA., Stanciu, LA., Synthesis of CeO₂-based core/shell nanoparticles with high oxygen storage capacity; *International Nano Letters*, 2017, 7:187-193.

Author's Response to Decision Letter for (RSOS-181861.R0)

See Appendix B.

Decision letter (RSOS-181861.R1)

14-Jan-2019

Dear Dr Luong:

Title: Structure and catalytic behavior of CuO-CeO₂ prepared by High-Energy Ball Milling
Manuscript ID: RSOS-181861.R1

It is a pleasure to accept your manuscript in its current form for publication in Royal Society Open Science. The chemistry content of Royal Society Open Science is published in collaboration with the Royal Society of Chemistry.

Yours sincerely,
Dr Laura Smith
Publishing Editor, Journals
Royal Society of Chemistry
Thomas Graham House
Science Park, Milton Road
Cambridge, CB4 0WF
Royal Society Open Science - Chemistry Editorial Office

RSC Associate Editor
Comments to the Author:
(There are no comments.)

Reviewer(s)' Comments to Author:

Appendix A

Reviewer's comments to article:

"Structure and catalytic behavior of CuO-CeO₂ prepared by High-Energy Ball Milling"

This is an interesting article focusing on the preparation of CuO-CeO₂ catalytic materials which are of important in the fabrication of The three way catalysts (TWCs). The CuO-CeO₂ composite powders were prepared at different ration of CuO to CeO₂ and different high energy ball milling time. The fabrication and testing methods are highly reliability. The results of this research show higher potential of total oxygen storage capacity of CuO-CeO₂ composites in comparison to traditional CeO₂-ZrO₂ materials.

I suggest that this article is suitable to publish in the Journal of Royal Society Open Science after the minor revisions as following:

1. There are several English mistakes through the article. The authors should take a proofreading.
2. Fig. 1 and Fig. 9 need to be edited to for clear reading: the axis titles and tick labels were overlapped
3. There are several mistakes in reference section that need to be corrected following the format of the journal and also, for citation through the text in the article.

Appendix B

Dear Editors

Thank you very much for your considerate email and sending the invaluable comments that the reviewers proposed on our paper which could be publishing in the Journal of Royal Society Open Science, entitled “Structure and catalytic behavior of CuO-CeO₂ prepared by High-Energy Ball Milling”. In our opinion, all the suggestions made by the referees would improve the manuscript, and we have carefully followed their indications. Our point-by-point answers are as follows:

Comments from the reviewer 1 and our response:

This is an interesting article focusing on the preparation of CuO-CeO₂ catalytic materials which are of important in the fabrication of the three way catalysts (TWCs). The CuO-CeO₂ composite powders were prepared at different ration of CuO to CeO₂ and different high energy ball milling time. The fabrication and testing methods are highly reliability. The results of this research show higher potential of total oxygen storage capacity of CuO-CeO₂ composites in comparison to traditional CeO₂-ZrO₂ materials. I suggest that this article is suitable to publish in the Journal of Royal Society Open Science after the minor revisions as following:

1. There are several English mistakes through the article. The authors should take a proofreading.

We revised carefully our manuscript, thank you for your comment

2. Fig. 1 and Fig. 9 need to be edited to for clear reading: the axis titles and tick labels were overlapped 3. There are several mistakes in reference section that need to be corrected following the format of the journal and also, for citation through the text in the article.

Figure 1 and Figure 2 were revised to make them more clearly; the mistakes in reference section were corrected, thank you.

Comments from the reviewer 2 and our response:

1. In line 56: the authors stated that “Although not detected, some CuO phase may exist as nanostructured or amorphous state.” There is no experimental indication of this statement. The authors should confirm this statement experimentally.

Thank you for your comment, based on our experience about ball milling and referred some papers, it could appear nanostructured or amorphous state due to the milling ex. E. R. Leite, L. P. S. Santos, N. L. V. Carreno, and E. Longo, Appl. Phys. Lett. 78, 2148 (2001), C.C.Koch, Nanostructured Materials, Volume 9, Issues 1–8, 1997, Pages 13-22, so in this paper we suggested this state .

2. There is no stability work for CeO₂-CuO system in the manuscript. It is known that although the presence of CuO besides CeO₂ enhances the OSC values compared to CeO₂-ZrO₂ or CeO₂-TiO₂, the thermal stability of the CuO-containing samples is quite low. The authors should discuss this in their manuscript.

This manuscript, we focus on the new method to prepare mixed CeO₂-CuO by high-energy mechanical milling, the reduction of CuO was observed when milling was processed in air, the thermal stability of the CuO was also discussed on Fig.8. Thank you.

3. The total OSC of CeO₂-CuO system is not directly related to the structural oxygen defects present in CeO₂ system. It is also related to the high reducibility of the CuO phase. If the authors calculated theoretical the OSC values, they would also realize this effect.

In this study, The total OSC improved which was relative the change crystallite structure of CuO and CeO₂ due to the milling, beside that it is also related to the high reducibility of the CuO phase. We will calculate and compare total OSC of CeO₂-CuO system which prepares ball milling with other method ex sol-gel in other paper. Thank you for your comments.

4. The authors did not use XPS and Raman Spec in the study to show the effect of mechanical milling on the density of structural oxygen defects in metal oxide particle system. The results obtained from the mentioned experimental methods will shed light on the enhanced OSC property and improve the author's results.

Thank you for your comments. We tried using XPS to observe the effect of mechanical milling on structural oxygen defects in metal oxide particle system (the XPS results will be attached in this revised manuscript). However, it showed that, the state of CuO is not stability, the CuO is oxidized easily due to samples treatment, the reason that crystallite structure of CuO was changed due to the milling, so it is difficult to use this method for calculation structural oxygen defects.

5. Some of the beneficial references are missing in the manuscript. The following papers can be used to improve the statements and address some of the concerns listed above. Therefore, they should be cited in the manuscript.

Uzunoglu, A., Zhang, H., Andreescu, S., Stanciu, A. L., CeO₂-MO_x (M: Zr, Ti, Cu) mixed metal oxides with enhanced oxygen storage capacity, *Journal of Materials Science* (2015) 50: 3750-3762, DOI 10.1007/s10853-015-8939-7.

Uzunoglu, A., Kose, DA., Stanciu, LA., Synthesis of CeO₂-based core/shell nanoparticles with high oxygen storage capacity; *International Nano Letters*, 2017, 7:187-193.

The references are now formatted. Thank you.

We believe that the revised paper includes all the important comments and corrections, which would, we hope, suffice the publication standard of the Royal Society Open Science Journal.

Yours sincerely,

Nguyen The Luong (on behalf of the authors)